# The First Two Years of COVID-19 Hospitalization Characteristics and Costs: Results from the National Discharge Registry

**DOI:** 10.3390/healthcare12100958

**Published:** 2024-05-07

**Authors:** Pierpaolo Ferrante

**Affiliations:** Department of Occupational and Environmental Medicine, Epidemiology and Hygiene, Italian National Workers’ Compensation Authority (INAIL), Via Stefano Gradi 55, 00143 Rome, Italy; p.ferrante@inail.it; Tel.: +39-06-5487-2733

**Keywords:** healthcare costs, COVID-19 costs, COVID-19 hospitalizations, COVID-19 hospital length of stay

## Abstract

Background: The COVID-19 pandemic has emerged as the primary global health challenge of the new millennium. Understanding its impact on health systems and learning from these experiences are crucial for improving system resilience against future health crises. This paper examines hospitalizations related to COVID-19 in Italy from 2020 to 2021, with a specific focus on the costs associated with these admissions. Design and methods: This is a retrospective, population-based study of Italian hospitalizations of patients diagnosed with COVID-19 during the 2020–2021 period, using data extracted from the National Hospital Discharge Registry. The outcome variables considered include hospital admissions, costs, and length of stay. Results: In Italy, hospitalizations for COVID-19 totaled 357,354 in 2020 and 399,043 in 2021, with the transfer rate being three times higher than that of other patients. Hospitalizations were predominantly concentrated in the northern regions, especially during the first year. Mortality rates increased with age, while hospitalization rates peaked in the youngest and oldest age groups. The financial impact of COVID-19 hospitalizations was approximately €3.1 billion in 2020 and €3.6 billion in 2021. The cost per admission was around €8000 for standard care and €24,000 for intensive therapy in both years. Conclusion: Conducting a cost-benefit analysis of implementing a protective pad around the entire health system, which leverages networks of family doctors and nurses connected in real-time, could be an important step in strengthening health system resilience.

## 1. Introduction

Severe acute respiratory syndrome coronavirus 2 (SARS-CoV-2) was first identified in late 2019 in the city of Wuhan (Hubei, China) [1]. It causes the coronavirus disease 2019 (COVID-19), with symptoms ranging from mild, flu-like symptoms to severe pneumonia [2]. The World Health Organization (WHO) declared the outbreak a Public Health Emergency of International Concern from 30 January 2020 to 5 May 2023, and a pandemic on 11 March 2020 [3]. The virus quickly spread around the world, and during the first two years of the pandemic (January 2020 to December 2021), weekly WHO data reported 280,614,414 confirmed cases and 5,446,891 deaths (https://data.who.int/dashboards/covid19/data?n=c, accessed on 1 March 2024). One of the most significant social effects of the pandemic was its profound impact on hospital capacities worldwide [4]. In 2020–2021, the European Centre for Disease Prevention and Control (ECDC) reported 12,614,410 COVID-19 hospitalizations in Austria, Belgium, Bulgaria, Cyprus, Czechia, Estonia, France, Ireland, Liechtenstein, Lithuania, Luxembourg, the Netherlands, Romania, Slovakia, Slovenia, and Spain (https://www.ecdc.europa.eu/en/publications-data/download-data-hospital-and-icu-admission-rates-and-current-occupancy-covid-19, accessed on 1 March 2024). During the same period in the US, the Centers for Disease Control and Prevention reported 3,719,818 hospitalizations across the entire country (https://covid.cdc.gov/covid-data-tracker/#trends_weeklyhospitaladmissions_select_00, accessed on 1 March 2024). Due to the complexity of treating COVID-19 patients, especially in intensive care units, hospitalization costs have increased worldwide. A systematic review found that in Germany, the total cost of hospitalization in intensive care per patient was USD 100,789 [5]. Even though the world was neither prepared nor coordinated for such an emergency, there was an impressive mobilization of human societies worldwide. The lethality of the virus of the original strain was quickly estimated using the infection fatality ratio (IFR), assessed through several national serosurveys [6,7]. By the end of 2020, global medicines agencies had conditionally approved several vaccines based on different technologies, with others close behind [8,9]. Most governments established unprecedented public health policies, including social distancing, remote working, and lockdowns, to reduce the spread of the virus [10]. Furthermore, there was an impressive proliferation of mathematical models aimed at predicting and managing the pandemic, as well as evaluating the implemented policies [11]. Italy was the first European country to be hit by the new virus and declared a health emergency status from 31 January 2020 to 31 March 2022. The unexpectedly high speed of transmission quickly led to hospital saturation and forced the government to implement a national lockdown [12]. The hardest period was during the first two years of the pandemic, before the vaccine campaign (which started in the last days of 2020) began to have the desired effects. The immunization program was delayed due to fears of possible vaccination-related side effects in a non-negligible proportion of the population, thereby increasing hospitalization costs and sparking discussions on related ethical issues that monopolized public opinion [13]. To enhance global healthcare systems by learning from the COVID-19 experience, it is crucial to analyze both hospitalization characteristics and the associated financial burdens resulting from the pandemic. Although hospitalization costs for COVID-19 patients have been estimated worldwide, national comparisons are impractical due to the use of different methods for calculating these costs [5,14]. However, understanding the costs of hospitalizations within a country can improve preparedness for future challenges by enhancing hospital resilience and ensuring appropriate compensation for exposed workers, including healthcare professionals. Globally, several countries have classified the pandemic as an occupational disease or a work injury under specific conditions when contracted in the workplace. Following recommendations from EU commissions [15], nine national workers’ compensation authorities in Europe have acknowledged SARS-CoV-2 infection in their existing recognition schemes without necessitating regulatory changes [16]. By equating the virulence of the virus to a violent incident, Italy has allowed for COVID-19 infection to be recognized as a work injury [17].

This article is part of a broader project focused on assessing hospital characteristics associated with principal occupational respiratory diseases. It specifically aims to describe the COVID-19 hospitalizations in Italy and, for the first time, estimate the associated costs with these cases.

## 2. Methods

### 2.1. Settings

In the Italian public health system, the regional administrations have significant autonomy in managing and organizing healthcare services within their territories. They are typically responsible for covering hospital charges by allocating the funds received from the central government to hospitals based on Diagnosis-Related Group (DRG) and Major Diagnostic Categories (MDC) coding systems. The central administration maintains functions related to the coordination and control of the provided service at a national level. The Ministry of Health establishes the National Standard Hospital Charges (NSHC), representing the maximum costs applicable to hospitalizations grouped by DRG (in cases of acute care) and MDC (in cases of rehabilitation or long-term care cases). For each group of admission, it also establishes a length of stay threshold (*thr*) beyond which the daily cost changes. Based on whether or not patients were admitted to intensive care, an additional cost specifically for COVID-19 hospitalizations was established by a decree on 12 August 2021.

### 2.2. Participants

This study included all patients in Italy diagnosed with COVID-19, and their corresponding hospitalization episodes, during the years 2020 and 2021. The study period includes all hospitalization data available at the date requested.

### 2.3. Outcomes

The primary outcomes included the annual number of COVID-19 hospitalizations, along with the associated length of hospital stay and costs. Secondary outcomes were the characteristics of both the hospitals and the patients.

### 2.4. Data Sources/Measurement

The Ministry of Health provided two aggregated datasets from the National Discharge Registry for years 2020–2021. The first dataset includes the number of patients by demographic, i.e., year, gender, and age group (0 years; 1–4 years; 5–13 years; 14 years; 15–24 years; 25–44 years; 45–64 years; 65–74 years; 75+ years) and their residence at the regional level. The second dataset expands on this by including the number of hospital admissions and length of stay, categorized by the same demographic variables and further detailed by primary medical treatment (coded by ICD-9-CM), Diagnosis-Related Groups (DRGs, version 24) with distinctions between medical and surgical types, care activity (including pregnancy-related, acute care, long-term care, and rehabilitation), hospitalization regimen (ordinary or day-care), and outcomes at discharge (deceased, transferred, or discharged to residence). For analytical purposes, we combined the age groups 5–13 and 14 years into a single group (5–14 years) and, according to the National Institute of Statistics (ISTAT), we grouped the 20 regions into 5 macro-areas, North-West (Aosta Valley, Piedmont, Lombardy, and Liguria), North-East (Trentino-Alto Adige, Friuli-Venezia Giulia, Emilia-Romagna, Veneto), Center (Tuscany, Marche, Umbria, and Lazio), South (Abruzzo, Basilicata, Molise, Apulia, Campania, Calabria), and Major Islands (Sicily and Sardinia). From the annual discharge reports of the Ministry of Health for 2020 and 2021, we obtained the total number of hospitalizations, the length of stay in days, and the total estimated costs [13,14]. Following cost estimates in those reports, we applied the NSHCs as defined by the Ministry’s decree of 12 October 2012 and *thr*s, as specified in the decree of 18 December 2008. Excluding admissions to day hospitals with medical DRGs, NSHCs for acute care are cumulatively expressed for the entire period up to *thr* time and as a daily charge for any period beyond *thr*. For rehabilitation, long-term care, and day hospital admissions in acute care settings with medical DRGs, NSHCs are set on a daily basis, distinguishing between periods up to and exceeding *thr*. Since we lack a variable indicating admission to intensive therapy, we have used ICD-9-CM treatment codes, specifically 96.7 (other continuous invasive mechanical ventilation), as a proxy for an Intensive care stay. We downloaded the Italian population by age and macro-area from the site of the ISTAT (https://demo.istat.it/app/?l=it&a=2020&i=POS, accessed on 1 March 2024). Statistics about all Italian hospitalizations (number of admissions, length of stay, and estimated costs) were taken from the Ministry of Health reports [18,19].

#### Statistical Analysis

We began by aggregating episodes of hospitalization that resulted in death at discharge, based on patient characteristics. This aggregated data was then merged with the existing patient database to incorporate the number of deceased patients. Within this enhanced dataset, we analyzed the patients’ characteristics by assessing the annual rate of unique hospitalized patients per 1000 residents (1000 patients/population), segmented by age and macro-area. Since the hospitalization data for 2020 begins on 20 February, the annual rate was obtained by rescaling the patient-to-population ratio using the factor 366/316. Additionally, we calculated the percentages of deaths and of males (Table 1). For the hospitalization dataset specifically, we examined the number of admissions and the total days of hospital stay, along with the mean duration per admission and number of transfers. All these metrics were then compared with those of other hospitalizations (Table 2). Cost assessment was conducted in several steps using the aggregated dataset, which includes the number of admissions (*n*) and days of stay (*GG*), segmented by the previously mentioned variables. Each record was associated to the corresponding *thr* and NSHC, both within (ch_1_) and beyond (ch_2_) them. We then calculated the cumulative threshold (*T*) for the grouped admissions as:*T* = *thr* × *n*

Costs of group admissions in records related to acute care, except those to day hospitals with medical DRGs (ACC), were then assessed as
*ACC* = *ch*_1_ × *n* + min{0,*GG*-*T*}× *ch*_2_
where min{0,*GG*-*T*} represents the number of days exceeding the threshold for the entire group of admissions. Costs of group admissions in records related to rehabilitation and long-term care, as well as acute care to day hospitals with medical DRGs costs (RLC), were calculated as:*RLC* = min{*GG*,*T*} × *ch*_1_ + min{0,*GG*-*T*}× *ch*_2_
where min{*GG*,*T*} represents the cumulative days for the entire group of admissions if it is lesser than the cumulative threshold; otherwise, it is equal to the cumulative threshold. We also added an extra cost of €3713 × *n* to all groups of admissions that were not transferred and €9697 × *n* to all groups of admissions with principal treatments coded as 96.7 in ICD-9-CM. Since the general expenses in the reports [13,14] are calculated net of the extra costs attributed to COVID-19, we added to them our estimates. Finally, 2020 costs were deflated by means of annual consumer price indexes provided by the Italian National Institute of Statistics. From the total costs, we derived the costs per admission and per day by simply dividing by the total number of admissions and days, respectively (Table 3). Finally, we compared all the previously mentioned metrics between cases that underwent continuous invasive mechanical ventilation and those that did not (Table 4).

## 3. Results

### 3.1. Patients’ Characteristics

During the initial two years of the COVID-19 pandemic (2020–2021), there were 300,336 hospitalized patients in 2020 and 335,287 in 2021. Mortality rates increased with age, remaining below 3% for individuals under 44 years but rising exponentially to 40% in those aged 75 and older. Hospitalization rates by age showed a U-shaped pattern, with peaks in infants and the elderly, with the lowest rates being in the 5–24-year age group (Table 1). The median age of patients discharged alive was below 70 years, while it was nearly 80 years for those who died in the hospital. Notably, the highest values for both outcomes were observed in the North (see Appendix A). Although gender differences were slight, females predominated in the 15–44 age group, while males were more prevalent in other age groups. The northern regions experienced a significant concentration of cases, especially in the first year. The hospitalization rate in Northern Italy was twice that of the rest of the country (Table 1), with Lombardy being the most affected region (refer to Appendix A).

### 3.2. Hospitalizations Characteristics

The high number of transfers increased the total number of hospitalizations attributed to COVID-19 to 357,354 in 2020 and 399,043 in 2021, representing 5.6% and 5.8% of all admissions, respectively. Patients admitted for COVID-19 experienced a transfer rate three times higher (16–17% vs. 5.5–5.7%) than those hospitalized for other reasons. Excluding transfers, COVID-19 admissions and their relative percentages dropped to 4.8% and 5.1% for 2020 and 2021, respectively. The total days of hospital stay for COVID-19 patients were 4,918,162 in 2020 (10% of the total) and 5,911,253 in 2021 (12% of the total). The mean length of stay was 13.8 days in 2020 and 14.8 days in 2021 for COVID-19 hospitalizations, compared to 7.1 and 6.8 days for other discharges, respectively (Table 2). COVID-19 hospitalizations were almost all in acute care (94% in both years), only a small part underwent surgical treatments (6.7% in 2020 and 8.4% in 2021), and those sent to day hospitals almost tripled in 2021, from 1.1% to 2.9% (Appendix A).

### 3.3. Hospitalization Costs and Intensive Care

The estimated total cost for COVID-19 hospitalizations in 2021 in euros was €3,131,200,175, in 2020 (€8762 per admission and €637 per day) and €3,605,043,360 in 2021 (€9034 per admission and €610 per day). Without considering the extra financial burden attributed to COVID-19, the hospitalization cost would have been €1,985,178,226 in 2020 (€5555 per admission and €404 per day) and €2,306,040,144 in 2021 (€5779 per admission and €390 per day) (Table 3). COVID-19 hospitalizations with other continuous invasive mechanical ventilation as the principal treatment were €18,143 (5.1%) in 2020 and €21,255 (5.3%) in 2021. The corresponding mean lengths of stay were 17.5 and 19.7 days, respectively. Compared to all the other hospitalizations, the length of stay increased by an average of 4 days in 2020 and by 5 days in 2021. Hospitalization costs for patients with invasive mechanical ventilation amounted to €429,114,405 (€23,652 per admission and €1354 per day) in 2020 and €512,940,599 (€24,133 per admission and €1225 per day) in 2021 (Table 4). The extra financial burden increased the cost of these admissions by about 66%. For the other COVID-19 hospitalizations, the total costs were €2,702,085,770 in 2020 (€7966 per admission and €587 per day) and €3,092,102,761 in 2021, with costs per admission increasing to €8185 and costs per day decreasing to €563, respectively. The extra financial burden increased the cost of these admissions by about 55%.

## 4. Discussion

The recent COVID-19 pandemic has highlighted global unpreparedness for managing the consequences of the emergence of a new virus with a high transmission rate. During the peaks of infection waves, many national health systems were pushed to the brink of collapse, and extreme measures to reduce contagion, such as lockdowns, proved to be the only effective countermeasures. These measures significantly impacted economic growth and public health worldwide [20,21]. In Italy, the gross domestic product collapsed by 7.5% in 2020, but the following year it rebounded by +9.7% (https://www.istat.it/it/archivio/288173, accessed on 1 March 2024). By describing significant hospital characteristics and estimating hospitalization costs, this paper can provide decision-makers with a detailed picture of the pandemic from the hospital network’s point of view.

According to the territorial spread of the virus, the distribution of hospitalized patients by residence exhibited a South-to-North gradient that correlated with levels of industrialization and mobility. At the beginning of 2020, the virus likely entered the country through Milan’s airport system (the capital city of the Lombardy region), which includes one intercontinental and two international airports (one of which is in Bergamo—the most severely affected Italian city). Subsequently, the virus spread according to the origin-destination matrix of goods and food transportation and population mobility [22]. The national lockdown in mid-March helped prevent the virus from further penetrating the central and southern regions. From the second wave, the virus spread throughout the country, influenced by local mobility flows (Table 1 in the main document and Appendix A).

Although it was expected that disease severity would increase with age (Appendix A), the increased hospital admission rates of infants compared to older children may be attributed to several factors. For example, it could result from heightened protective measures implemented for infants compared to children aged 1–14. Additionally, infants typically have closer contact with their parents than children aged 1–4 years, potentially making them more susceptible to household transmission. The social behavior in Italian families could at least partially explain the higher prevalence of females in the 25–44-year age group, as mothers usually spend more time on childcare than fathers [23]. Demographic-related patterns indicate that conducting widespread testing campaigns and rigorous contact tracing efforts significantly contribute to monitoring the virus’s epidemiology and slowing its transmission by obstructing the virus’s preferred routes of spread.

It is important to highlight the high percentage of patient transfers, presumably because the initial hospitals were overwhelmed. Especially in the first months of the pandemic, hospitals in high-virus areas faced a tsunami of emerging infections. Without efficient primary care, emergency rooms in hospitals became the frontline against the virus and many patients were subsequently distributed among hospitals as a second step. A consequence has been overexposing health workers to the risk of infection, forcing local management to reorganize the service [24]. Healthcare and social workers faced a high burden, with a percentage of complaints at work reaching 62% of the nearly 200,000 total recorded during the study period [17]. To alleviate the pressure on emergency rooms, measures such as involving family doctors as the first point of contact for potential infections and implementing improved patient triage protocols were adopted [25]. The implemented countermeasures yielded significant reductions in both hospital transfers and healthcare worker complaints during the second year. Hospital transfer rates fell from 17.4% to 15.9% (Table 1), while healthcare worker complaints decreased from 67.2% to 52.3% [17,26].

The high number of hospital admissions (5.6% of the total) was accompanied by a hospital stay double that of non-COVID-19 patients (representing 11% of the total). In addition, more than 5% of cases required mechanical ventilation, strongly increasing the complexity of treatment. The estimated hospital costs for the first year of the pandemic (which covered the 10-month period from March to December 2020) exceeded 3 billion euros. In the second year, these costs rose to 3.5 billion euros (Table 3). Admissions to intensive care tripled the cost per hospitalization in both years, reaching nearly 24,000 euros, and more than doubled the daily cost—from 587 to 1354 euros in 2020 and from 563 to 1225 euros in 2021 (Table 4). Although these figures likely underestimate the actual costs, as they do not account for expenses related to keeping available hospital beds and emergency room services for COVID-19 patients (as defined by a decree on 12 August 2021), they still accounted for up to 13% of the total hospitalization costs and 2% of the gross domestic product.

The broad impact of the crisis on people, encompassing personal health (both mental and physical), behavioral changes (such as social distancing, remote working, and distance learning), and economic challenges (stemming from lockdowns and reduced mobility), has deeply affected societies worldwide. This forces us to reflect on our past mistakes to avoid repeating them. Strengthening primary healthcare could significantly enhance health system resilience in the face of new challenges. Additionally, it is worth assessing the benefits and costs of implementing a protective pad around the entire health system. This would involve a network of family doctors and nurses, connected in real time to the health system, tasked with monitoring the local epidemiological situation, protecting vulnerable individuals through telehealth visits, and establishing a triage system for infections in their initial phases. Such a network could effectively manage the spread of viruses and protect hospitals from unpredictable waves of admissions.

### Limitations

This paper has two main limitations. Firstly, we categorized hospitalizations with the ICD-9-CM treatment code of 93.9 as admissions to intensive care, which represent approximately 5% of total hospitalizations instead of the expected 10% [18]. Secondly, we did not account for the costs associated with waiting for hospital beds for acute COVID-19 patients nor the costs related to waiting in emergency departments for the management of confirmed and suspected COVID-19 cases.

## 5. Conclusions

The COVID-19 pandemic highlighted critical vulnerabilities in the Italian healthcare system. Driven by the need for a comprehensive approach that goes beyond immediate healthcare needs, this study analyzed the financial and operational aspects of hospitals’ response to the pandemic. For the first time, it estimated the substantial financial burden of COVID-19 hospital admissions on healthcare budgets and GDP. This further underscored the system strains, evident in the tripled transfer rate compared to other diseases and in the high number of complaints by healthcare workers. However, promising solutions emerged during the second year. The involvement of family doctors as the first point of contact for potential infections along with safer triage protocols were associated with significantly reduction in hospital transfers (1.5%) and healthcare worker complaints (15%). These reductions have the potential to not only improve staff well-being but also lead to a more efficient use of healthcare resources. Investing in adaptable healthcare strategies and empowering primary care with a robust network of family doctors and nurses can safeguard against future crises and ensure a healthcare system that is both responsive and preventive.

## Figures and Tables

**Table 1 healthcare-12-00958-t001:** COVID-19 hospitalized patient characteristics.

Macro-Area ^1^	Age ^2^	2020			2021		
		Admission Rate	Deaths (%)	Male (%)	Admission Rate	Deaths (%)	Male (%)
North-West							
	75+	32.4	40.1	51.2	21.2	31.1	49.0
	65–74	16.3	23.7	65.5	11.6	16.4	60.9
	45–64	7.5	8.4	67.3	5.2	5.4	64.7
	25–44	2.5	1.8	49.2	2.1	0.9	44.8
	15–24	0.9	0.8	47.4	0.8	0.6	45.7
	5–14	0.9	0.4	56.2	0.7	0.2	58.9
	1–4	2.7	0.1	55.1	3.1	0.1	56.4
	0	16.5	0.1	52.8	16.3	0.2	53.8
North-East							
	75+	27.8	33.9	48.3	23.8	31.0	48.2
	65–74	11.1	18.2	63.7	11.1	15.5	61.0
	45–64	4.9	6.5	65.3	5.3	5.4	64.8
	25–44	1.9	1.5	49.0	2.0	1.3	51.0
	15–24	0.8	0.4	48.8	0.7	0.6	48.1
	5–14	0.6	0.4	60.3	0.5	0.8	62.6
	1–4	1.8	0.4	60.4	1.9	0.3	57.4
	0	9.2	0.5	56.7	11.3	0.2	51.8
Center							
	75+	16.7	34.4	48.6	17.7	31.1	47.8
	65–74	8.2	18.9	63.4	10.7	15.8	58.3
	45–64	4.2	7.7	64.5	6.1	6.0	62.8
	25–44	2.0	1.7	45.9	3.2	1.1	46.6
	15–24	0.8	1.0	47.1	1.2	0.4	47.1
	5–14	0.7	0.2	56.6	1.1	0.2	54.6
	1–4	2.4	0.1	54.4	3.4	0.1	53.5
	0	11.2	0.1	54.8	16.4	0.2	54.3
South							
	75+	9.6	39.8	51.4	11.5	40.1	49.6
	65–74	5.6	25.1	65.2	7.7	25.5	61.1
	45–64	2.9	11.0	67.7	4.0	11.9	63.5
	25–44	1.2	3.3	50.8	1.7	2.9	46.1
	15–24	0.5	1.7	50.4	0.6	1.3	44.9
	5–14	0.4	0.9	57.9	0.4	0.6	54.9
	1–4	2.3	0.2	54.0	3.0	0.0	56.3
	0	7.9	0.6	56.1	8.9	0.0	54.9
Major Islands							
	75+	8.3	34.2	47.7	11.1	36.5	47.2
	65–74	4.5	21.8	61.8	6.6	21.8	58.5
	45–64	2.3	9.4	64.9	3.6	10.1	61.9
	25–44	1.2	2.0	39.6	1.6	1.6	44.5
	15–24	0.6	0.5	41.8	0.8	1.0	42.6
	5–14	0.8	0.3	53.8	1.0	0.5	54.1
	1–4	2.9	0.2	55.3	3.9	0.0	57.7
	0	6.3	0.0	55.8	9.7	0.2	57.0

^1^ there are 3275 patients with foreign residence. ^2^ there are 44 patients with missing data on age.

**Table 2 healthcare-12-00958-t002:** Number of hospitalizations and days of length of stay and transfers for COVID-19 and non-COVID-19 admissions.

Year	COVID-19				Non-COVID-19			
	Discharges	Total Days	Days per Hospitalization	Transfers	Discharges	Total Days	Days per Hospitalization	Transfers
2020	357,354	4,918,162	13.8	62,126	6,151,486	43,852,606	7.1	348,117
2021	399,043	5,911,253	14.8	63,361	6,622,507	44,985,112	6.8	354,714

**Table 3 healthcare-12-00958-t003:** Total, per admission and per day, hospitalization costs for COVID-19 and all diagnoses.

Year	COVID-19 Costs			All Admissions Costs		
	Total	x Admission	x Day	Total	x Admission	x Day
2020	3,131,200,175	8762	637	25,430,213,596	3907	521
2021	3,605,043,360	9034	610	27,616,154,755	3933	543

**Table 4 healthcare-12-00958-t004:** Total, per admission and per day, hospitalization costs for COVID-19 admissions in intensive care.

Year	Mechanical Ventilation (ICD-9: 93.7)	Admissions	Days	Mean Length	Cost		
					Total	x Admission	x Day
2020							
	No	339,211	4,601,316	13.6	2,702,085,770	7966	587
	Yes	18,143	316,846	17.5	429,114,405	23,652	1354
2021							
	No	377,788	5,492,388	14.5	3,092,102,761	8185	563
	Yes	21,255	418,865	19.7	512,940,599	24,133	1225

## Data Availability

Data are contained within the article and Appendix A.

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
