# Peer review of "The First Two Years of COVID-19 Hospitalization Characteristics and Costs: Results from the National Discharge Registry"

_healthcare, 2024, doi:10.3390/healthcare12100958_

Round 1

Reviewer 1 Report

Comments and Suggestions for Authors

 There are few issues that need to be addressed by authors before this manuscript could be reconsidered for publication:

(1) In Keywords, the first letter of each group of words should be capitalized.

(2) The main conclusion of this paper is that implementing a protective pad around the entire health system that leverages networks of family doctors and nurses, connected in real time to the entire health system.However, in the analysis of formulas and tables, the data of medical consumption after joining the network of family doctors and nurses and consumption without joining the network are not compared. How can the conclusion be drawn?

Comments on the Quality of English Language

Moderate editing of English language required.

Author Response

Introduction

Firstly I would like to thank reviewers for taking the time to read my paper and providing insightful comments that significantly helped improve it. Specifically,

  • I have corrected some errors in the R code related to the costs estimate,
  • I have adjusted the 2020 hospitalization rate to account for the entire year rather than just a 10-month period (table 1),
  • I have added the median age of patients by year (table 2 of supplemental materials),
  • I have refined the main conclusion to better reflect the obtained results,
  • I have rebalanced the abstract,
  • I have thoroughly discussed all the reported results.
  • I have added several references

All the changes I have done are in red text.

Reviewer 1:

There are few issues that need to be addressed by authors before this manuscript could be reconsidered for publication:

  1. In Keywords, the first letter of each group of words should be capitalized.

Response

Done. I have changed them an used the capital letters.

  1. The main conclusion of this paper is that implementing a protective pad around the entire health system that leverages networks of family doctors and nurses, connected in real time to the entire health system. However, in the analysis of formulas and tables, the data of medical consumption after joining the network of family doctors and nurses and consumption without joining the network are not compared. How can the conclusion be drawn?

Response

Thank you for your valuable comment. Since my results can only suggest performing a cost-benefit analysis, I have changed the sentence of the abstract’s conclusion to:

“Conducting a cost-benefit analysis of implementing a protective pad around the entire health system, which leverages networks of family doctors and nurses connected in real-time, can be a crucial step in strengthening health system resilience. “

As well as the last paragraph in the discussion to

“The broad impact of the crisis on people, encompassing personal health (both mental and physical), behavioral changes (such as social distancing, remote working, and distance learning), and economic challenges (including activity reduction), has deeply affected societies worldwide. This forces us to reflect on our past mistakes to avoid repeating them. Strengthening primary healthcare could significantly enhance health system resilience in the face of new challenges. Additionally, it is worth assessing the benefits and costs of implementing a protective pad around the entire health system. This would involve a network of family doctors and nurses, connected in real time to the health system, tasked with monitoring the local epidemiological situation, protecting vulnerable individuals through telehealth visits, and establishing a triage system for infections in their initial phases. Such a network could effectively manage the spread of viruses and protect hospitals from unpredictable waves of admissions.”

Reviewer 2 Report

Comments and Suggestions for Authors

Dear authors,
I enjoyed reviewing your paper titled: "The first two years of COVID-19 hospitalization characteristics and costs: results from the national discharge registry." The work is compelling and presents excellent insights. However, I would like to highlight some aspects that could enhance it.

The abstract would benefit from a more balanced space allocation among its various sections. For instance, the background section appears scant and superficial compared to the disproportionately extended results section. Please consider making the abstract more concise, balanced, and impactful.
Regarding keywords, certain elements, such as healthcare costs, should be incorporated into the paper.

While the introduction is captivating, the argumentation on vaccinations needs more depth regarding the ethical considerations that significantly impacted Italy, potentially affecting costs (please see DOI: 10.3390/vaccines10101602 ). Additionally, there are some typographical errors, and several citations are missing (e.g., in the phrase: "The unexpected high speed of transmission...").

The discussion section of the introduction should emphasize the need for further research and clearly outline the study's objectives. Furthermore, the correlation with occupational aspects is only introduced towards the end of the introduction; please elaborate on them within the body of the introduction. Specifically, the discussion section, the discussion line "Hospitalization costs for COVID-19, etc." could enrich this introductory passage.

Section 2.1 would benefit from improved flow and specificity; please ensure it is comprehensive yet accessible.

Section 2.2 provides a detailed explanation of the study period, considering that 2020 was not entirely impacted by the COVID-19 pandemic, potentially compromising the reliability of comparisons.

Section 2.4 outlines the subdivision of Italian regions into groups; please specify the individual groups and assess whether regional variations occurred.

The data in the statistical analysis section should align consistently with the section's content. Additionally, ensure that the formulas used are adequately explained and accessible. It should be clarified whether the study received approval from an economist and a statistician.

In the results section, consider reevaluating the data in light of the date-related issues mentioned. Similarly, review the discussion section based on the provided information.

Tables 1, 3, and 4 require clarification. Please revise them and reassess the clarity of the tables and the organization of data within them.

Author Response

Introduction

Firstly I would like to thank reviewers for taking the time to read my paper and providing insightful comments that significantly helped improve it. Specifically,

  • I have corrected some errors in the R code related to the costs estimate,
  • I have adjusted the 2020 hospitalization rate to account for the entire year rather than just a 10-month period (table 1),
  • I have added the median age of patients by year (table 2 of supplemental materials),
  • I have refined the main conclusion to better reflect the obtained results,
  • I have rebalanced the abstract,
  • I have thoroughly discussed all the reported results.
  • I have added several references

All the changes I have done are in red text.

Reviewer 2:

I enjoyed reviewing your paper titled: "The first two years of COVID-19 hospitalization characteristics and costs: results from the national discharge registry." The work is compelling and presents excellent insights. However, I would like to highlight some aspects that could enhance it.

  1. The abstract would benefit from a more balanced space allocation among its various sections. For instance, the background section appears scant and superficial compared to the disproportionately extended results section. Please consider making the abstract more concise, balanced, and impactful.

Response

Thank you for your suggestion. I rebalanced the abstract, reduced the number of words to 234 and tried to make it more impactful.

  1. Regarding keywords, certain elements, such as healthcare costs, should be incorporated into the paper.

Response

Done

  1. While the introduction is captivating, the argumentation on vaccinations needs more depth regarding the ethical considerations that significantly impacted Italy, potentially affecting costs (please see DOI: 10.3390/vaccines10101602 ).

Response

After the sentence

The hardest period was during the first two years of pandemic, before the vaccination campaign (started in last days of 2020) began to have the desired effects.

I added the following sentence:

The immunization program was delayed due to fears of possible vaccination-related side effects in a non-negligible proportion of the population, thereby increasing hospitalizations costs and sparking discussions on related ethical issues that monopolized public opinion [14]

  1. Additionally, there are some typographical errors, and several citations are missing (e.g., in the phrase: "The unexpected high speed of transmission...").

Response

The introduction was revised sentence by sentence. In the highlighted case, the sentence was rewritten as follows:

The unexpectedly high speed of transmission quickly led to hospital saturation and forced the government to implement a national lockdown (12).

  1. The discussion section of the introduction should emphasize the need for further research and clearly outline the study's objectives. Furthermore, the correlation with occupational aspects is only introduced towards the end of the introduction; please elaborate on them within the body of the introduction. Specifically, the discussion section, the discussion line "Hospitalization costs for COVID-19, etc." could enrich this introductory passage.

Response

The discussion section of the introduction was rewritten as follows:

To enhance global healthcare systems by learning from the COVID-19 experience, it is crucial to analyze both hospitalizations characteristics and the associated financial burdens resulting from the pandemic. Although hospitalization costs for COVID-19 patients have been estimated worldwide, national comparisons are impractical due to the use of different methods for calculating these costs (5,17). However, understanding the costs of hospitalizations within a country can improve preparedness for future challenges by enhancing hospital resilience and ensuring appropriate compensation for exposed workers, including healthcare professionals. Globally, several countries have classified the pandemic as an occupational disease or a work injury under specific conditions when contracted in the workplace. Following recommendations from EU commissions [15], nine national workers' compensation authorities in Europe have acknowledged SARS-CoV-2 infection in their existing recognition schemes without necessitating regulatory changes [16]. By equating the virulence of the virus to a violent incident, Italy has allowed for COVID-19 infection to be recognized as a work injury [17].

  1. Section 2.1 would benefit from improved flow and specificity; please ensure it is comprehensive yet accessible.

Response

I have changed several sentences of the subsection. It should now fit your requests.

“In the Italian public health system, the regional administrations have significant autonomy in managing and organizing healthcare services within their territories. They are typically responsible for covering hospital charges by allocating the funds received from the central government to hospitals based on diagnosis-related group (DRG) and Major Diagnostic Categories (MDC) coding systems. The central administration maintains functions related to the coordination and the control of the provided service at national level. By the Ministry of Health, it establishes the National Standard Hospital Charges (NSHC), representing the maximum costs applicable to hospitalizations grouped by DRG (in cases of acute care) and MDC (in cases of rehabilitation or long-term care cases). For each group of admission, it also establishes a length of stay threshold (thr) beyond which the daily cost changes. Based on whether or not patients were admitted to intensive care, an additional cost specific for COVID-19 hospitalizations, was established by a decree on August 12, 2021.”

  1. Section 2.2 provides a detailed explanation of the study period, considering that 2020 was not entirely impacted by the COVID-19 pandemic, potentially compromising the reliability of comparisons.

Response

Thank you for pointing out this issue and give me the opportunity to address it as follows

·         In section 2.2: I have added the sentence “The study period includes all hospitalization data available at the date request.”

  • I have rescaled the rate of hospitalized patients by the factor 366/316 and explained it in the “statistical analysis” subsection by the following sentence:

“Since the hospitalization data for 2020 begins on February 20, the annual rate was obtained by rescaling the patient-to-population ratio using the factor of 366/316 (Table 1)”

  • I have thought about rescaling also the total number of admissions of and the total costs related to 2020, however the aim of this paper is not to compare years but to provide an estimate of the actual hospitalization costs which are still unknown in Italy. The only comparisons between years present in the document are the average length of stay per hospitalization and the percentage of transfers, which do not require adjustments on an annual basis.

  1. Section 2.4 outlines the subdivision of Italian regions into groups; please specify the individual groups and assess whether regional variations occurred.

Response

I replaced the original sentence with
“For analytical purposes, we combined the age groups 5-13 and 14 years into a single group (5-14 years) and, according to the National Institute of Statistics (ISTAT), we groped the 20 regions into macro-area 5 areas, such as North-West (Aosta Valley, Piedmont, Lombardy and Liguria), North-East (Trentino-Alto Adige, Friuli-Venezia Giulia, Emilia-Romagna, Veneto), Centre (Tuscany, Marche, Umbria and Lazio), South ( Abruzzo, Basilicata, Molise, Apulia,  Campania, Calabria), and major Islands (Sicily and Sardinia).”

I have also included the following table in supplemental materials with the regional metrics. Additionally, I have added commentary on the territorial distribution of infections, emphasizing Lombardy's primary role in the pandemic.

Table 2 Population, COVID-19 hospitalization rates and percentage of hospital deaths and males by region and year. Italy 2020-2021

macro-area

Region

2020

2021

Patients1

hosp. rate2

deaths (%)

Male (%)

Patients1

hosp. rate2

deaths (%)

Male (%)

Northwest

Piedmont

32,073

8.6

27.6

57.2

28,862

6.8

20.4

55.4

Aosta Valley

1,063

9.8

24.6

53.0

1,071

8.6

10.0

50.8

Lombardy

82,259

9.5

23.8

58.9

59,871

6.0

17.1

55.7

Liguria

12,185

9.3

25.5

55.0

12,465

8.2

15.4

54.3

Northeast

Trentino-Alto Adige

6,660

7.2

18.0

56.5

6,906

6.4

13.5

55.6

Veneto

22,505

5.3

22.4

56.6

24,729

5.1

18.9

56.6

Friuli-Venezia Giulia

5,608

5.4

22.4

55.1

9,235

7.7

22.2

56.4

Emilia-Romagna

34,733

9.0

22.1

53.9

35,188

7.9

17.9

54.0

Centre

Tuscany

14,823

4.6

19.3

54.9

21,403

5.8

16.2

55.3

Umbria

2,915

3.9

20.5

56.3

4,986

5.8

17.0

54.7

Marche

6,163

4.7

26.9

55.8

8,100

5.4

20.1

55.7

Lazio

27,827

5.6

18.4

55.4

42,116

7.3

14.1

53.6

South

Abruzzo

5,326

4.8

22.8

58.1

7,343

5.7

17.4

56.3

Molise

717

2.8

25.8

59.0

1,133

3.8

28.9

58.3

Campania

12,908

2.6

24.1

61.1

19,628

3.5

24.6

55.1

Apulia

12,836

3.8

20.0

57.9

18,601

4.7

21.0

57.2

Basilicata

1,097

2.3

23.7

55.9

1,992

3.7

19.5

55.3

Calabria

2,256

1.4

16.8

58.1

5,082

2.7

19.4

56.8

Major Islands

Sicily

11,645

2.8

18.1

54.0

20,432

4.2

19.3

53.7

Sardinia

3,301

2.4

19.7

55.7

4,305

2.7

18.9

54.8

  1. The data in the statistical analysis section should align consistently with the section's content.

Response

Additionally, ensure that the formulas used are adequately explained and accessible. It should be clarified whether the study received approval from an economist and a statistician.

Response

Thank you for pointing out this issue. I have corrected errors in the formulas and provided clearer explanations. The rationale behind this approach is as follows: since each record represents a group of admissions, the National Standard Hospital Charge should be multiplied by the total number of admissions (n) if it refers to a period. However, if it pertains to a single day, the charge should be multiplied by the number of days (GG before T and GG-T beyond it)..

I have changed the original sentences as follows:

“Cost assessment was conducted in several steps using the aggregated dataset, which includes the number of admissions (n) and days of stay (GG), segmented by the previously mentioned variables. Each record was associated to the corresponding thr and NSHC within (ch1) and beyond (ch2) it. We then calculated the cumulative threshold (T) for the grouped admissions as:T = thr × nCosts of group admissions in records related to acute care, except those to day hospital with medical DRGs, (ACC)  were then assessed asACC = ch1 × n + min{0,GG-Tch2where min{0,GG-T} represents the number of days exceeding the threshold for the entire group of admissions. Costs of group admissions in records related to rehabilitation and long-term care as well as acute care to day hospital with medical DRGs (RLC) costs were calculated as:RLC = min{GG,T} × ch1 + min{0,GG-Tch2

where min{GG,T} represents the cumulative days for the entire group of admissions if it is lesser than the cumulative threshold; otherwise, it is equal to the cumulative threshold.“

I am a statistician and I have already used this approach in other papers (https://bmjopen.bmj.com/content/11/8/e046456, https://link.springer.com/article/10.1007/s00420-020-01637-z), the only difference is that in this case the hospitalization data are aggregated

  1. In the results section, consider reevaluating the data in light of the date-related issues mentioned. Similarly, review the discussion section based on the provided information.

Response

I have adjusted the hospitalization rate for 2020 by rescaling the hospitalized patient-to-population ratio with the factor 366/316, as explained in the 'Statistical Analysis' subsection. The relevant sentence reads:

“Since the hospitalization data for 2020 begins on February 20, the annual rate was obtained by rescaling the patient-to-population ratio using the factor of 366/316 (Table 1).”

While I considered rescaling the total number of admissions and related costs for 2020, the primary aim of this paper is not to compare different years but to provide an estimate of the actual hospitalization costs, which remain still unknown in Italy. The only comparisons between years present in the document are the average length of stay per hospitalization and the percentage of transfers, which do not require adjustments on an annual basis.

  1. Tables 1, 3, and 4 require clarification. Please revise them and reassess the clarity of the tables and the organization of data within them.

Response

I have added the description of the tables in the subsection “Statistical analysis” of the methods section as follows

Within this enhanced dataset, we analyzed the patients’ characteristics by assessing the annual rate of unique hospitalized patients per 1.000 residents (1.000 patients/population), segmented by age, and Territorial division. Additionally, we calculated the percentages of deaths and of male patients. Since the hospitalization data for 2020 begins on February 20, the annual rate was obtained by rescaling the patient-to-population ratio using the factor of 366/315 (table 1). For the hospitalization dataset specifically, we examined the number of admissions and the total days of hospital stay, along with mean duration per admission (days/admissions) and these metrics were then compared with those of other hospitalizations (table 2). From the total costs, we derived the costs per admission and per day by simply dividing by the total number of admissions and days, respectively (table 3). Finally, we compared all the previously mentioned metrics between cases that underwent continuous invasive mechanical ventilation and those that did not (table 4).

Reviewer 3 Report

Comments and Suggestions for Authors

Dear authors, 

Thank you for presenting the results of your research! Your article is interesting and the results are important for the public health! 

As you porbably know the median is more appropriate to present the lenght of hospital stay. However you work with aggregated data so the mean is the only option.

It would be useful if you present the costs also as % of GDP in order to be comparable with other countries.  

Comments on the Quality of English Language

Minor English editing is needed. For example in Table 1 there are Nord (Italian word instead of North)-West and Nord-East regions. 

Author Response

Introduction

Firstly I would like to thank reviewers for taking the time to read my paper and providing insightful comments that significantly helped improve it. Specifically,

  • I have corrected some errors in the R code related to the costs estimate,
  • I have adjusted the 2020 hospitalization rate to account for the entire year rather than just a 10-month period (table 1),
  • I have added the median age of patients by year (table 2 of supplemental materials),
  • I have refined the main conclusion to better reflect the obtained results,
  • I have rebalanced the abstract,
  • I have thoroughly discussed all the reported results.
  • I have added several references

All the changes I have done are in red text.

Reviewer 3:

Thank you for presenting the results of your research! Your article is interesting and the results are important for the public health! 

Response

Thank you very much for your kind words. They are a wonderful acknowledgment of my efforts.

  1. As you probably know the median is more appropriate to present the length of hospital stay. However you work with aggregated data so the mean is the only option.

Response

Your considerations are correct; however, your comments have provided me with an intriguing insight. In supplemental materials, I have added the following table which contains the median age of patients by year and macro-area

Table 1: Medin age of patients and population by macro-area and year

Year

macro-area

Median Age (years)

Hospitalizations outcome

Population

alive

deaths

2020

Nord-west

67.7

78.0

48.0

Nord-east

70.4

78.6

47.8

Centre

66.2

78.0

47.9

South

61.5

76.7

45.5

Major Islands

61.3

77.0

46.4

Overall

67.3

78.0

47.2

2021

Nord-west

67.6

78.1

48.2

Nord-east

68.5

78.4

48.0

Centre

62.2

77.7

48.2

South

60.9

76.1

46.0

Major Islands

61.0

76.8

46.9

Overall

65.8

77.7

47.5

 I commented it in the subsection” Patients’ characteristics” of results with the following sentence:The median age of patients discharged alive was below 70 years, while it was nearly 80 years for those who died in the hospital. Notably, the highest values for both outcomes were observed in the north (see Table 1 of the supplemental materials)

  1. It would be useful if you present the costs also as % of GDP in order to be comparable with other countries.  

Response

Done. I have added in the following paragraph in the “discussion” section:

The high number of hospital admissions (5.6% of the total) was accompanied by a hospital stay double that of non-covid-19 patients (representing 11% of total). In addition, more than 5% of cases required the mechanical ventilation strongly increasing the complexity of treatment. The estimated hospital costs for the first year of the pandemic (which covered the 10-month period from March to December 2020) exceeded 3 billion euros. In the second year, these costs rose to 3.5 billion euros.  Admissions to intensive care tripled the cost per hospitalization in both years, reaching nearly 24,000 euros, and more than doubled the daily cost—from 587 to 1,354 euros in 2020 and from 563 to 1,225 euros in 2021. Although these figures likely underestimate the actual costs, as they do not account for expenses related to waiting for hospital beds and emergency room services for COVID-19 patients (as defined by a decree on August 12, 2021), they still accounted for up to 13% of the total hospitalization costs and 2% of the gross domestic product.  I have also added some other reference of the economical impact in discussion with the following 2 sentences -       These measures significantly impacted economic growth and public health worldwide (20,21). In Italy the gross domestic product collapsed by 7.5% in 2020, but the following year it rebounded by +9.7%[1].-       The broad impact of the crisis on people, encompassing personal health (both mental and physical), behavioral changes (such as social distancing, remote working, and distance learning), and economic challenges (stemming from lockdowns and reduced mobility), has deeply affected societies worldwide

[1] https://www.istat.it/it/archivio/288173

Reviewer 4 Report

Comments and Suggestions for Authors

(1)   The manuscript could use additional proofreading. There were a few typos and occasional unusual word choices.

(2)   The authors should provide a basis discussion of the Italian healthcare system for those not familiar with it. Does the government cover the entire cost or do citizens pay a certain percentage of the cost?

(3)   The authors provide descriptive statistics but do not discuss the implications of their findings. Are there any conclusions we can draw from the facts that Northern Italy had a higher hospitalization rate compared to the rest of the country? Are there political, social, or demographic reasons that might explain this outcome? Why did females primarily dominate the 15 to 44 age group?

(4)   While the authors need to do a better job linking their results to the discussion section.

(5)   There needs to be a conclusion section that provides discussion on what this research adds to the literature on COVID-19 and preparedness.

Comments on the Quality of English Language

Overall, there were minimal concerns regarding the quality of English used in the submitted paper. There were a few small typos and a couple of strange word choices but a quick proofread should address these issues.

Author Response

Introduction

Firstly I would like to thank reviewers for taking the time to read my paper and providing insightful comments that significantly helped improve it. Specifically,

  • I have corrected some errors in the R code related to the costs estimate,
  • I have adjusted the 2020 hospitalization rate to account for the entire year rather than just a 10-month period (table 1),
  • I have added the median age of patients by year (table 2 of supplemental materials),
  • I have refined the main conclusion to better reflect the obtained results,
  • I have rebalanced the abstract,
  • I have thoroughly discussed all the reported results.
  • I have added several references

All the changes I have done are in red text.

Reviewer 4:

The manuscript could use additional proofreading. There were a few typos and occasional unusual word choices.

  1. The authors should provide a basis discussion of the Italian healthcare system for those not familiar with it. Does the government cover the entire cost or do citizens pay a certain percentage of the cost?
  2.  

Response

The cost was typically paid by the national health system at regional level. I changed the original sentence to

“In the Italian public health system, the regional administrations have significant autonomy in managing and organizing healthcare services within their territories. They are typically responsible for covering hospital charges by allocating the funds received from the central government to hospitals based on diagnosis-related group (DRG) and Major Diagnostic Categories (MDC) coding systems. The central administration maintains functions related to the coordination and the control of the provided service at national level. By the Ministry of Health, it establishes the National Standard Hospital Charges (NSHC), representing the maximum costs applicable to hospitalizations grouped by DRG (in cases of acute care) and MDC (in cases of rehabilitation or long-term care cases)”

  1. The authors provide descriptive statistics but do not discuss the implications of their findings. Are there any conclusions we can draw from the facts that Northern Italy had a higher hospitalization rate compared to the rest of the country? Are there political, social, or demographic reasons that might explain this outcome? Why did females primarily dominate the 15 to 44 age group?

Thank you for your suggestions. I commented your points as follows:

According to the territorial spread of the virus, the distribution of hospitalized patients by residence exhibited a south-to-north gradient that correlated with levels of industrialization and mobility. At the beginning of 2020, the virus likely entered the country through Milan's airport system (the capital city of the Lombardy region) which includes one intercontinental and two international airports (one of which is in Bergamo—the most severely affected Italian city). Subsequently, the virus spread according to the origin-destination matrix of goods and food transportation and population mobility [22]. The national lockdown in mid-March helped prevent the virus from further penetrating the central and southern regions. From the second wave, the virus spread throughout the country, influenced by local mobility flows (Table 1 in the main document and in the supplemental materials)

  1. While the authors need to do a better job linking their results to the discussion section, there needs to be a conclusion section that provides discussion on what this research adds to the literature on COVID-19 and preparedness.

Response

Thank you for highlighting these points. This paper provides, for the first time, estimates of the costs associated with COVID-19-related hospitalizations in Italy. It also underscores the necessity of conducting a cost-benefit analysis to strengthen the primary healthcare system via a supportive network of family doctors and nurses. In response to your comments, I have made the following additions:

  • Discussion Section: I have included a new paragraph that connects our findings with broader impacts and implications:

The broad impact of the crisis on people, encompassing personal health (both mental and physical), behavioral changes (such as social distancing, remote working, and distance learning), and economic challenges (stemming from lockdowns and reduced mobility), has deeply affected societies worldwide. This forces us to reflect on our past mistakes to avoid repeating them. Strengthening primary healthcare could significantly enhance health system resilience in the face of new challenges. Additionally, it is worth assessing the benefits and costs of implementing a protective pad around the entire health system. This would involve a network of family doctors and nurses, connected in real time to the health system, tasked with monitoring the local epidemiological situation, protecting vulnerable individuals through telehealth visits, and establishing a triage system for infections in their initial phases. Such a network could effectively manage the spread of viruses and protect hospitals from unpredictable waves of admissions. 

-   Conclusion Section: I have added a conclusion section that summarizes our findings and their implications

The COVID-19 pandemic has strained healthcare systems financially and operationally. The estimated direct costs associated with Italian hospital admissions and intensive care have placed a significant burden on healthcare budgets, consuming a notable percentage of the gross domestic product. Moreover, the broader impacts on personal health, societal behaviour, and economic stability calls for an integrated response that extends beyond immediate healthcare needs. To this end, strengthening primary healthcare systems through a supportive network of family doctors and nurses offers pathway worth investigating. By learning from the lessons of this pandemic and investing in comprehensive and adaptive healthcare strategies, we can safeguard against future crises, thereby ensuring that our health systems are not only responsive but also preventive.

Round 2

Reviewer 1 Report

Comments and Suggestions for Authors

The manuscript has been revised point to point. Now, it can be accepted.

Author Response

Thank you for your positive feedback

Reviewer 2 Report

Comments and Suggestions for Authors

Dear Authors,

I want to extend my warmest congratulations to you for your dedication to addressing the review comments and implementing revisions in the manuscript. The changes you made were comprehensive and greatly enhanced the clarity and robustness of your work. I appreciate your cooperation and carefully considering my feedback throughout the review process.

Thank you.

Author Response

Thank you for your careful review and support of this manuscript

Reviewer 4 Report

Comments and Suggestions for Authors

I appreciate the efforts taken by the authors to strengthen the manuscript. While I see a significant improvement in the manuscript from the original submission, the conclusion is still weak. The authors should strengthen this aspect of the paper. It needs to address, in greater detail, the findings from the study and what it has added to the literature on this topic.

Author Response

Introduction

I'm pleased to address reviewer 4's final point. To strengthen the conclusions and my advice on implementing protective measures, I've also added a sentence in the discussion highlighting the decreasing trend in healthcare worker complaints (changes marked in red).

Reviewer 4

  1. I appreciate the efforts taken by the authors to strengthen the manuscript. While I see a significant improvement in the manuscript from the original submission, the conclusion is still weak. The authors should strengthen this aspect of the paper. It needs to address, in greater detail, the findings from the study and what it has added to the literature on this topic.

Response

Thank you for your feedback on the conclusion. I've strengthened the conclusion by adding details on the study's findings and its contribution to the literature. This includes highlighting the system weaknesses revealed by the study (tripled transfer rate, worker complaints), the first-ever cost estimations for Italy, and the significant burden on healthcare budgets and the national economy:

“By studying the impact of the pandemic on the hospitals network, this paper highlighted the weakness of the system response, evidenced by a tripled transfer rate compared to the other diseases and the overwhelming number of complaints of health workers. Moreover, it has estimated, for the first time in Italy, the direct costs associated with hospital admissions and intensive care, highlighting the significant burden placed on healthcare budgets and its impact on the gross domestic product.”
